# Transcriptional Study of Radiofrequency Device Using Experimental Mouse Model

**DOI:** 10.3390/ijms26094460

**Published:** 2025-05-07

**Authors:** Xiaofeng Li, Zheng Wang, Xiaoman Li, Xiaofeng Fan, Xinyu Lu, Yanan Li, Yehua Pan, Ziyan Zhu, Mingxi Zhu, Wei Li, Leo Chan, Suyun Yu, Yanhong Pan, Yuanyuan Wu

**Affiliations:** 1Jiangsu Key Laboratory for Pharmacology and Safety Research of Chinese Materia Medica, School of Pharmacy, Nanjing University of Chinese Medicine, Nanjing 210023, China; iamleexxx@163.com (X.L.); wangzheng@njucm.edu.cn (Z.W.); lixm@njucm.edu.cn (X.L.); liyn0110@163.com (Y.L.); 20220813@njucm.edu.cn (Y.P.); 20220825@njucm.edu.cn (Z.Z.); 20220824@njucm.edu.cn (M.Z.); leochan1982@gmail.com (L.C.); 2School of Pharmacy, Nanjing University of Chinese Medicine, Nanjing 210023, China; 300619@njucm.edu.cn; 3School of Medicine, Nanjing University of Chinese Medicine, Nanjing 210023, China; 846107@njucm.edu.cn (X.L.); yusuyun@njucm.edu.cn (S.Y.); 4School of Integrated Chinese and Western Medicine, Nanjing University of Chinese Medicine, Nanjing 210023, China; weili91@njucm.edu.cn

**Keywords:** radiofrequency, collagen, matrix metalloproteinases, skin aging, Fos

## Abstract

Radiofrequencies have shown efficacy in addressing skin aging. Despite their effectiveness, few studies have explored how radiofrequencies affect the skin transcriptome. This study utilized mouse models divided into two age groups (four-month-old and one-year-old mice) to assess the impact of a radiofrequency device on skin collagen and elastin. A combination of histological analysis, Western blot analysis, real-time PCR and transcriptome sequencing was employed. Histological analysis revealed significant increases in dermis thickness and collagen fiber volume following radiofrequency treatment in both age groups. Quantitative PCR and Western blot analysis indicated that the levels of collagen-related genes and proteins were higher in the four-month-old group. Transcriptome sequencing identified 465 and 1867 differentially expressed genes (DEGs) in the skin of the 4-month-old mice and 1-year-old mice, respectively. GO and KEGG analyses elucidated the molecular mechanisms, revealing that the interleukin-17 and tumor necrosis factor signaling pathways may play crucial roles in collagen regeneration induced by radiofrequencies. Additionally, decreased expression of matrix metalloproteinase-9 and increased expression of the transcription factor Fos were identified as potential biomarkers of collagen regeneration. Immunofluorescence and immunohistochemistry staining demonstrated that radiofrequencies activate fibroblasts and inhibit macrophage alternative activation in the skin. This study identifies key genes and biological pathways involved in radiofrequency treatment and provides a foundation for a deeper understanding of the molecular mechanisms underlying collagen regeneration facilitated by radiofrequencies.

## 1. Introduction

As global life expectancy increases, aging presents significant challenges to human health and places a growing burden on healthcare systems. Among all organs, the skin, being the largest and most visible, shows the most perceptible signs of aging. Skin aging is a complex process that involves intrinsic and extrinsic mechanisms that lead to various structural and physiological changes [1]. Common manifestations of skin aging include sagging and wrinkles, primarily due to the loss and degradation of collagen. Collagen, the most abundant protein in mammals, is found extensively in the skin, tendons, bones and other tissues. From early adulthood, collagen levels decline at an approximate rate of 1.5% annually. To counteract this decline and enhance skin collagen, various treatments have been developed, including topical applications such as retinol and tretinoin, as well as advanced technologies like lasers, focused ultrasound and radiofrequencies [2]. Among these approaches, radiofrequency therapy has gained substantial clinical traction due to its user-friendly application protocol and favorable safety profile.

Radiofrequency technology is a non-invasive treatment that uses high-frequency electrical currents to generate heat energy within the skin tissues. This heat stimulates collagen regeneration and growth, thereby enhancing skin quality and appearance. Additionally, radiofrequency technology promotes improved blood circulation in the skin, contributing to increased radiance and elasticity. Clinically, several studies have demonstrated the efficacy of radiofrequency devices in reducing facial and body skin wrinkles [3,4,5]. In vivo research has confirmed their ability to enhance the density and structural integrity of collagen and elastic fibers. At the molecular level, radiofrequency-induced collagen production has been linked to elevated mRNA levels of transforming growth factor β (TGF-β) and vascular endothelial growth factor (VEGF). Moreover, radiofrequency energy has been shown to boost collagen and elastin synthesis through the up-regulation of NRF2/GLO-1 and modulation of M1/M2 macrophage polarization. It also inhibits the formation of advanced glycation end products (AGEs) and their receptor (RAGE), thereby reducing NF-κB activation and slowing collagen and elastin degradation [6]. Despite these insights, there remains a gap in understanding of the transcriptional mechanisms and the impact on the tissue microenvironment associated with radiofrequency treatment. For example, what are the biomarkers for the efficacy or sensitivity of radiofrequency treatment? What is the cellular crosstalk in the skin environment during the treatment? To address this, the present study employed RNA sequencing in mouse models to explore additional mechanisms underlying radiofrequencies’ effects on skin collagen production.

## 2. Results

### 2.1. Radiofrequencies Promote an Increase in Dermal Layer Thickness and Collagen Volume in Mice

Mice are a well-established model for aging research, offering detailed insights at both cellular and physiological levels. In this study, 4-month-old mice were used to represent the young adult phase, equivalent to a human age of approximately 20–30 years, while 1-year-old mice served as the middle-aged model, corresponding to a human age of about 38–47 years [7]. Figure 1A illustrates the experimental design, with 4-month-old and 1-year-old mice undergoing radiofrequency treatment. Histological analysis, including representative H&E and Masson’s trichrome staining of ICR mouse skin, is shown in Figure 1B,C. The dermis layers in the 4-month-old untreated mice were thicker compared to those in the 1-year-old untreated mice. Masson’s trichrome staining revealed a decrease in collagen bundle density and a looser arrangement in the older mice (Figure 1D). Following one week of radiofrequency treatment, there was a marked increase in dermal thickness and collagen bundle density in both the 4-month-old and 1-year-old mice (Figure 1B–E).

### 2.2. Radiofrequencies Promote Collagen Expression in Skin of 4-Month-Old Mice

In human skin, type I collagen makes up 80 to 90% of the total collagen, while type III constitutes 8 to 12% [8]. Elastin, consisting of crosslinked tropoelastin combined with microfibrils, forms elastic fibers that impart stretch and recoil to the skin [9]. In our study, we assessed the transcriptional and translational levels of type I collagen, type III collagen and elastin in mouse samples by qPCR and Western blot analyses. One week of radiofrequency treatment significantly increased the mRNA levels of the type I collagen-encoding gene *col1a1* and the type III collagen-encoding gene *col3a1* in the 4-month-old mice (Figure 2A–C), with corresponding up-regulation of Col1a1 and Col3a1 protein expression observed at the translational level (Figure 2D–G). In the 1-year-old mice, increases in average gene expression were noted at the protein level post-treatment, although no statistically significant differences were observed. These trends may reflect individual variability and warrant further investigation with larger sample sizes. Thus, radiofrequency treatment up-regulates collagen-related genes at both the transcriptional and translational levels in the skin of 4-month-old mice, while in 1-year-old mice, transcriptional levels of these genes are more responsive to treatment than translational levels.

### 2.3. Identification of Differentially Expressed Genes in the Skin Following Radiofrequency Treatment Using RNA Sequencing and Bioinformatic Analysis

To investigate the comprehensive gene expression profiles following radiofrequency treatment, bulk-tissue RNA sequencing (RNA-seq) was conducted on skin samples from 4-month-old and 1-year-old mice, both with and without radiofrequency treatment (*n* = 3 per group). Systematic differential gene expression (DEG) analysis was performed to identify genes associated with radiofrequency treatment. Using a cutoff value of |logFC| > 1.2 and *p*-values < 0.05, we identified 465 DEGs in the skin of 4-month-old mice and 1867 DEGs in the skin of 1-year-old mice. Volcano plots illustrating these DEGs are shown in Figure 3A,B. Significant genes meeting the criteria (*p*-value < 0.05, logFC > 1.2 and logFC < −1.2) are highlighted as red dots. Among these DEGs, 82 genes were common to the age groups and were associated with radiofrequency treatment. These 82 DEGs were analyzed using a heat map generated by the Heatmapper web server based on hierarchy, as depicted in Figure 3D.

To further explore the biological processes and molecular pathways related to skin remodeling following radiofrequency treatment, Gene Ontology (GO) analysis was conducted, including the categories of Biological Process (BP), Cellular Component (CC) and Molecular Function (MF) (Figure 3E). The GO analysis revealed that DEGs in both age groups were primarily involved in the regulation of body fluids, skin development, antimicrobial humoral response and water homeostasis in terms of Biological Processes. In the Cellular Component category, the most enriched DEGs were associated with organelles and the extracellular matrix (ECM). In the Molecular Function category, key DEGs were related to transcription factors and chemokines (Figure 3E). Additionally, to investigate the molecular pathways affected by radiofrequency treatment in both age groups, KEGG pathway analysis was performed. This analysis identified 20 significantly altered pathways, as shown in Figure 3F. Notably, the IL-17 signaling pathway, TNF signaling pathway and estrogen signaling pathway were among the most significantly altered pathways.

### 2.4. Protein–Protein Interactions (PPIs) Among DEGs Following Radiofrequency Treatment

The 82 DEGs were imported into the STRING database to construct a protein–protein interaction (PPI) network, as shown in Figure 4A. Ten hub genes—*MMP9*, *ITGAM*, *FOS*, *ATF3*, *SERPINB2*, *CCL20*, *CXCL5*, *NCF1*, *F9* and *FGFR1*—were identified within the PPI network using the Cytoscape (v3.10.3) plug-in CytoHubba (Figure 4B). To validate the RNA-seq results, *Mmp9* and *Fos*, which were significantly differentially expressed according to the RNA-seq analysis, were selected for qRT-PCR validation. The qRT-PCR results confirmed that the expression changes observed were consistent with those found in the RNA-seq analysis. Specifically, in both the 4-month-old and the 1-year-old groups, *Mmp9* levels decreased while *Fos* levels increased (Figure 5A,B).

### 2.5. Radiofrequency Treatment Activates Fibroblasts in the Dermis of the Skin

Dermal fibroblasts are the primary cells responsible for collagen production within the dermis. These fibroblasts are heterogeneous and can be categorized anatomically into papillary dermal fibroblasts and reticular dermal fibroblasts. Functionally, papillary fibroblasts are fundamental for the formation and growth of new hair follicles, whereas reticular fibroblasts, which exhibit higher levels of α-SMA expression, are involved in producing the ECM, including the collagen matrix, and contribute to skin repair and regeneration following wound injury [10,11]. Our results demonstrate that radiofrequency treatment increases the number of α-SMA^+^ cells (Figure 6A). This effect was more robust in the 1-year-old group compared to the 4-month-old group (Figure 6B).

### 2.6. M2 Macrophages Accumulate in the Skin of Aged Mice While Radiofrequency Treatment Reprograms Macrophage Activation in the Dermis

Macrophages collaborate with fibroblasts to maintain tissue homeostasis, including in the skin, by enhancing fibroblast activation and collagen secretion [12]. Accumulated evidence demonstrates that age-induced changes in macrophages may lead to a pro-inflammatory state characterized by an alternatively activated (M2-like) phenotype [13]. However, the impact of radiofrequency treatment on macrophage activation in the skin remains elusive. The CIBERSORT algorithm, which calculates the proportion of various immune cell types, was employed to investigate the immune cell composition and to study the pathogenesis of diseases. We utilized this algorithm to analyze immune cell distributions and to visualize the results using box plots for skin tissues from the 4-month-old and 1-year-old groups with or without radiofrequency treatment (Figure 7A,B). Macrophages accounted for the highest proportion (more than 20%) among all immune cell populations in this analysis, representing the predominant cellular subset. This immune cell infiltration analysis combined with immunohistochemical staining of CD206 revealed that M2 macrophages accumulate in the skin of aged mice, with a reduction in M2 macrophages following radiofrequency treatment in both age groups (Figure 7C,D).

## 3. Discussion

Radiofrequency technology is widely employed in medical aesthetics due to its anti-aging effects, which include reducing skin sagging and wrinkles while enhancing skin firmness, elasticity and radiance. Previous studies have shown that thermal damage caused by radiofrequency treatment initiates the skin’s wound repair process and stimulates collagen regeneration [14]. However, the molecular mechanisms and gene expression profiles underlying the anti-aging effects of radiofrequencies remain poorly understood. In this study, we observed significant differences in both histological and molecular characteristics of mouse skin following one week of radiofrequency treatment.

RNA-sequencing analysis of radiofrequency-treated versus untreated mouse skin provides a comprehensive list of significantly differentially expressed genes in the skin. Given the high similarities in histological, physiological and immunological properties between mouse and human skin, the molecular changes observed in our experimental animals are likely to reflect those in human skin treated with radiofrequencies in clinical settings. Besides histological alterations, mouse skin responds to the microthermal damage induced by radiofrequencies with distinct gene expression profiles. A limited number of overlapping differentially expressed genes in the 4-month-old and 1-year-old mice reveal unique gene expression patterns induced by radiofrequencies. These genes include damage response genes such as *Tnc*, *Itgam* and *Smtnl2*. *Tnc* and *Itgam* are directly involved in damage response, with *Tnc* focusing on tissue remodeling and inflammation and *Itgam* on immune cell function and response to injury. *Smtnl2*, while not as directly linked to damage response as the other two, still plays a role in the physiological processes that can be part of the body’s response to damage, particularly in smooth muscle tissues. We also screened out other genes, which include genes that regulate the cell cycle, DNA replication, cell survival and cytoskeletal remodeling. These genes are *Ly9*, *Ncf1*, *Fos*, *Aspn*, *Reg2*, *Treml1*, *Ccl20*, *Chrng*, *Atf3*, *Marchf7*, *Cstdc2*, *Lce1m*, *Cxcl5*, *Akap3*, *Acsm3* and *F9*. Many of these genes are also up-regulated during wound healing, suggesting that they represent a common transcriptional program involved in cellular stress and proliferation. However, these genes have not been shown to be associated with injury responses. Notably, GO and KEGG analyses highlighted the AP1 family transcription factor Fos. Previous studies demonstrated that mechanical-force-induced skin damage activates the EGFR-RAS-MAPK pathway, leading to Fos up-regulation that drives epidermal stem cell proliferation and renewal [15]. In our study, we observed consistent Fos up-regulation, suggesting its conserved role in the tissue remodeling process.

Furthermore, GO and KEGG analyses revealed significant activation of inflammatory responses, including the IL-17 signaling pathway and the TNF signaling pathways, which may stimulate cell proliferation and initiate skin regeneration. Chemokines, cytokines and proteolytic processing factors interact in inflammatory responses induced by microthermal injury. In this complex regulatory network, we observed changes in the activity of MMP genes involved in ECM remodeling following radiofrequency treatment. Mmp9, a matrix metalloproteinase, degrades collagen, gelatin, fibrin and other ECM components [16]. Differential expression analysis indicates that Mmp9 expression is down-regulated after radiofrequency treatment in both young and aged mice, which suggests a reduction potential in ECM degradation. Our findings underscore the critical involvement of inflammatory cascades in mediating cutaneous adaptive responses to radiofrequency exposure. Future investigation is needed to delineate the mechanisms through which controlled immune activation orchestrates tissue repair post-thermal injury, particularly focusing on the temporal regulation of pro-regenerative versus pro-fibrotic signaling pathways. Furthermore, the therapeutic potential of combinatorial regimens integrating anti-inflammatory immunomodulators with radiofrequency devices warrants systematic exploration in preclinical models.

Although histological analysis indicated significant differences between the young and aged groups, as well as between the untreated and the treated groups, the most pronounced molecular differences were observed in collagen-related genes and proteins within the young group (4-month-old mice). The observed discrepancy in radiofrequency efficacy between the 4-month-old group and the 1-year-old group may be attributed to the age-related alterations in the tissue microenvironment that collectively impair regenerative capacity. Mechanistically, aged skin exhibits (1) dysfunctional collagen deposition characterized by a fragmented matrix architecture that disrupts fibroblast adhesion and mechanotransduction signaling, (2) altered immune cell dynamics with increased M2-polarized macrophages that may dampen pro-regenerative crosstalk between fibroblasts and immune compartments [17], and (3) diminished expression of critical transcriptional regulators (e.g., AP1 factors) and chemokine gradients essential for orchestrating biomolecular cascades following thermal stimulation [18]. These degenerative changes are further exacerbated by age-associated adipose infiltration that physically disrupts dermal architecture [19]. Consequently, these alterations create a microenvironment less responsive to radiofrequency-induced tissue remodeling signals compared to younger skin with intact ECM integrity and robust paracrine signaling networks.

There are also limitations to our study. Although we identified Fos and MMP9 as the targets of radiofrequencies in skin aging treatment, validation of these targets using either genetic or pharmacologic manipulation has not yet been fulfilled. The age-related radiofrequency resistance due to alterations in the microenvironment still needs to be investigated in cohorts with large sample sizes and long-term observations, as well as radiofrequencies with more parameters. The detailed crosstalk among cells in skin ECM during tissue repair is worth exploring.

## 4. Conclusions

Taken together, the results demonstrate that radiofrequency-induced spatial heating in the dermal layer induces physical changes in the cellular microenvironment and triggers biomolecular cascades, such as IL-17 and TNF signaling pathways, to restore skin homeostasis. Additionally, Mmp9 and Fos have been identified as potential targets involved in skin remodeling (Figure 8). Therefore, this study provides biological insights into the use of radiofrequencies against skin aging and also provides experimental evidence supporting their use in future experimental and clinical applications.

## 5. Materials and Methods

### 5.1. Mice and Radiofrequency Treatment

ICR male mice, aged 6–8 weeks, were obtained from Shanghai SLAC Animal Co. (Shanghai, China) and were housed in the animal facility at the Nanjing University of Chinese Medicine, with a temperature of 24 °C as well as a 12 h/12 h light/dark cycle, with ad libitum access to food and water until 4 months of age and 1 year of age. The animals were monitored and cared for according to the guidelines set by the Animal Ethics Committee of Nanjing University of Chinese Medicine (approval number: 202209A069). The mice were randomly divided into two age groups upon their arrival at the animal facility: 4-month-old mice (*n* = 6) and 1-year-old mice (*n* = 5).

The animal experiments were designed as self-control studies in two age groups, where each subject served as its own control. This approach allowed for a direct comparison of effects within the same individual, thereby reducing variability and enhancing the reliability of the results. Prior to each treatment, the mice were anesthetized using isoflurane inhalation. Mouse dorsal skin was delineated into two 1 cm × 2 cm areas: one designated as untreated and the other as treated, following hair removal. The dorsal skin was cleaned and dried thoroughly. A gel (TriPollar^®^, Pollogen Ltd., Tel Aviv, Israel) was evenly applied to the designated treatment area. A multipolar radiofrequency device (model: TriPollar^®^ Stop VX2, Pollogen Ltd.) was utilized in low-range mode, delivering a power density of 8 W/cm^2^, with frequencies alternating between 0.9, 1.0 and 1.25 MHz. The dorsal skin was massaged until the integrated temperature indicator activated (target range: 40–42 °C), ensuring consistent application across the treated area. Treatments were conducted every other day for one week (a regimen optimized based on prior unpublished data showing peak mRNA profiles of collagen type I and type III and fibroblast activity at this interval; Figure 1A). Following the final treatment, the mice were euthanized via isoflurane inhalation (4% in oxygen) and cervical dislocation, and dorsal skin tissues were harvested for analysis.

### 5.2. Histopathological Analysis

Dorsal skin samples were fixed in 4% paraformaldehyde for 24 h, followed by embedding in paraffin and microtome sectioning. Sections were prepared with a 4 μm thickness and were stained with hematoxylin and eosin (H&E). To assess collagen fiber content, additional sections were stained with Masson’s trichrome. The stained sections were then imaged using the Mantra Pathology Workstation (PerkinElmer, Waltham, MA, USA). Dermal thickness and collagen fiber volume were quantified using ImageJ software (v1.8.0.112, Bethesda, MD, USA), based on measurements from the H&E and Masson’s trichrome staining, respectively.

### 5.3. Immunohistochemical (IHC) Staining

IHC staining was conducted following the manufacturer’s protocol (ZSGB-BIO, Beijing, China). Briefly, tissue sections were incubated overnight at 4 °C with the primary antibody CD206 (CST, 24595T, Danvers, MA, USA). This was followed by a 50 min incubation at room temperature with HRP-labeled Goat Anti-Rabbit IgG (H+L) (Bioworld, BS13278, Nanjing, China). Antigen–antibody complexes were visualized using diaminobenzidine (DAB). The stained sections were imaged using the Mantra Pathology Workstation (PerkinElmer, Waltham, MA, USA).

### 5.4. Immunofluorescence (IF) Staining

For tissue immunofluorescence staining, paraffin-embedded skin sections fixed with 4% paraformaldehyde (PFA) were deparaffinized and dehydrated through a series of xylene and ethanol washes. Antigen retrieval was performed in sodium citrate buffer. After blocking with 5% BSA for 30 min at room temperature, sections were incubated overnight at 4 °C with the primary antibody α-SMA (CST, 19245T, Danvers, MA, USA). Following this, the sections were incubated with the secondary antibodies for 1 h at room temperature. Sections were counterstained with Hoechst33342 to label nuclei. Images were captured from three random fields for each sample using a Zeiss Axiovert A1 microscope (Oberkochen, Germany).

### 5.5. Quantitative Real-Time Polymerase Chain Reaction (qPCR)

Dorsal skin tissues were snap-frozen with liquid nitrogen and homogenized with a micro tissue homogenizer (Servicebio, KIZ-III-F, Wuhan, China). Total RNA was isolated using TRIzol (Invitrogen/Life Technologies, Carlsbad, CA, USA) following the manufacturer’s protocol. The extracted RNA was quantified by NanoDrop (Thermo Fisher Scientific, ND-ONE-W, Waltham, MA, USA) and converted to cDNA using a reverse transcription kit (Vazyme, R223, Nanjing, China). The primers used are listed in Table 1, and qPCR was performed on an Applied Biosystems 7900HT (Applied Biosystems/Life Technologies, Foster City, CA, USA). Glyceraldehyde-3-phosphate dehydrogenase (Gaphd)-normalized mRNA for specific gene expression was calculated by the 2^−ΔΔCt^ method.

### 5.6. Western Blot Analysis

Protein samples from mouse dorsal skin were homogenized using a micro tissue homogenizer and then lysed with RIPA buffer, following the manufacturer’s instructions (Yeasen, 20115ES60, Shanghai, China). Protein concentration was determined with a BCA protein assay kit (Beyotime, P0012, Shanghai, China) according to the manufacturer’s instructions. For Western blotting, 30 μg of protein per sample was loaded on a sodium dodecyl sulfate–polyacrylamide (SDS) gel and separated by electrophoresis. The proteins were then transferred to polyvinylidene fluoride (PVDF) membranes using the wet transfer system (Bio-Rad, MiniProtean 3 Cell, San Francisco, CA, USA). The PVDF membranes were blocked with 5% non-fat milk in Tris-buffered saline and Tween 20 (TBST) and then incubated overnight at 4 °C with primary antibodies against type I collagen (Proteintech, 66761-1-Ig, Wuhan, China), type III collagen (Proteintech, 22734-1-AP, Wuhan, China) and elastin (Immunoway, YN2891, Suzhou, China), diluted in TBST containing 0.02% sodium azide and 5% BSA. After washing 4 times with TBST, the membrane was incubated with horseradish peroxidase (HRP)-conjugated secondary antibody (Bioworld Technology, BS13278, BS12478, Nanjing, China). The protein blot was visualized using enhanced chemiluminescence (ECL) reagents (Biosharp, BL520B, Hefei, China) and captured with a Bio-Rad imaging system (Bio-Rad, XRS+, San Francisco, CA, USA).

### 5.7. Image Analysis

The pathological tissue images and Western blot images were analyzed using ImageJ software in a blind manner. Specifically, all images were randomized and labeled by an independent researcher prior to analysis, ensuring that the analyst was unaware of the treatment groups. This approach minimized bias during the quantification of relevant parameters, allowing for objective evaluation of the histological features. Image analysis was performed with ImageJ software (v1.8.0.112, Bethesda, MD, USA).

### 5.8. RNA Sequencing

Sample selection and data exclusion: Samples were selected based on pathological phenotype and RNA stability, with three biological replicates per group included in the analysis. The samples from the same three biological replicates per group were subsequently used for molecular experiments (Western blot and qPCR analyses) to ensure consistency. No additional samples or data points were excluded beyond this selection criteria.

Tissue collection: Mouse skin tissues (about 100 mg per biological replicate) were harvested from each experimental group, immediately snap-frozen in liquid nitrogen and stored at −80 °C until processing.

RNA extraction: Total RNA from untreated and treated skin tissues was extracted using Trizol (Invitrogen, Carlsbad, CA, USA), following the manufacturer’s protocol, with 3 replicates per group according to the manufacturer’s instructions. RNA quantity and integrity was measured using the Nanodrop One (Thermo Fisher Scientific, MA, USA) and the Agilent 2100 Bioanalyzer (Agilent Technologies, PaloAlto, CA, USA), respectively. RNA degradation was examined by RNase-free agarose gel electrophoresis.

RNA sequencing: The mRNA was enriched by oligo(dT) magnetic beads, cut into short fragments and reverse transcribed into the first-strand cDNA using the First Strand Synthesis Reaction Buffer and Random Primer Mix (2×). The second-strand cDNAs were synthesized using the Second Strand Synthesis Reaction Buffer (10×) and Second Strand Synthesis Enzyme Mix. The library was constructed using the NEB Next Ultra RNA Library Prep Kit for Illumina (neb# 7530, New England Biolabs, Ipswich, MA, USA). Subsequently, the second-strand cDNAs were purified with AMPure XP Beads. The ends were repaired, poly(A) was added and the cDNA was ligated to Illumina sequencing adapters. The ligation products were purified using AMPure XP Beads (1.0×X), washed with 80% ethanol and eluted with ddH_2_O. The connecting products were used for PCR amplification. The cDNA library was constructed and sequenced on the Illumina sequencing platform (Illumina Novaseq6000) supplied by Gene Denovo Biotechnology Co. (Guangzhou, China).

Gene expression levels were normalized by the FPKM (fragments per thousand base transcripts/million map readouts) method, and the differential expression genes were screened with a fold change of ≥1.2 and a *p*-value < 0.05. Volcano maps and heat maps were generated using online website tools (https://www.omicsmart.com/), and GO enrichment and KEGG pathway analyses were generated using the clusterProfiler R package (v4.0) with a significance threshold of *p* < 0.05. Abundances of immune cell infiltration were analyzed by the Cibersort algorithm. Two-way ANOVA analysis was used to assess the differences in the proportions of immune cells. Results with *p* < 0.05 were considered significant. The RNA-sequencing data are available in the Gene Expression Omnibus (GEO) database under GSE278079.

### 5.9. Statistical Analysis

The paired Student’s *t*-test was used to assess significance for all data sets. GraphPad 8.02 software (version 8.02; San Diego, CA, USA) analyzed and processed the data results. Quantitative analysis of histological experiments was performed with either five or six biological replicates, as indicated individually. Data sets were analyzed with three biological replicates in the Western blot and qPCR experiments. Data were expressed as means ± standard deviations. *p* < 0.05 indicated a statistically significant difference between groups.

## Figures and Tables

**Figure 1 ijms-26-04460-f001:**
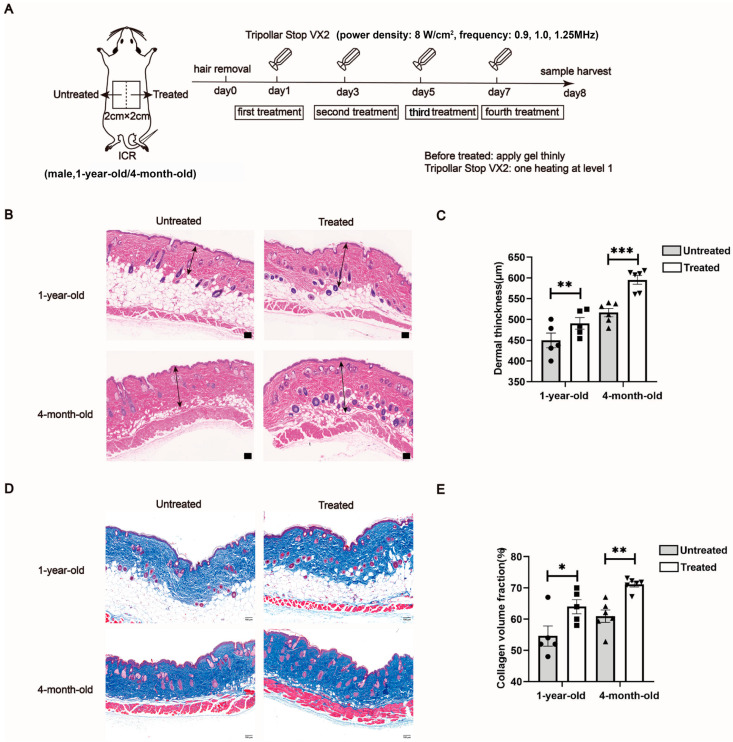
Effects of radiofrequency treatment on dermal thickness and collagen volume in mice of different ages. (**A**) Illustration of treatment schedule. (**B**,**C**) The histopathological change in dermal thickness was shown by H&E staining. The dermal thickness was measured using ImageJ. The arrows in (**A**) indicate the dermal layer. Scale bars = 100 μm. *n* = 5 in 1-year-old group, *n* = 6 in 4-month-old group. (**D**,**E**) The histology and morphological structure of mouse skin tissues were measured by Masson’s trichrome staining for labeling of collagen fibers in blue. Scale bars = 100 μm. *n* = 5 in 1-year-old, *n* = 6 in 4-month-old. Statistical test used in (**C**,**E**) is paired Student’s *t*-test. * *p* < 0.05, ** *p* < 0.01 and *** *p* < 0.001 versus untreated.

**Figure 2 ijms-26-04460-f002:**
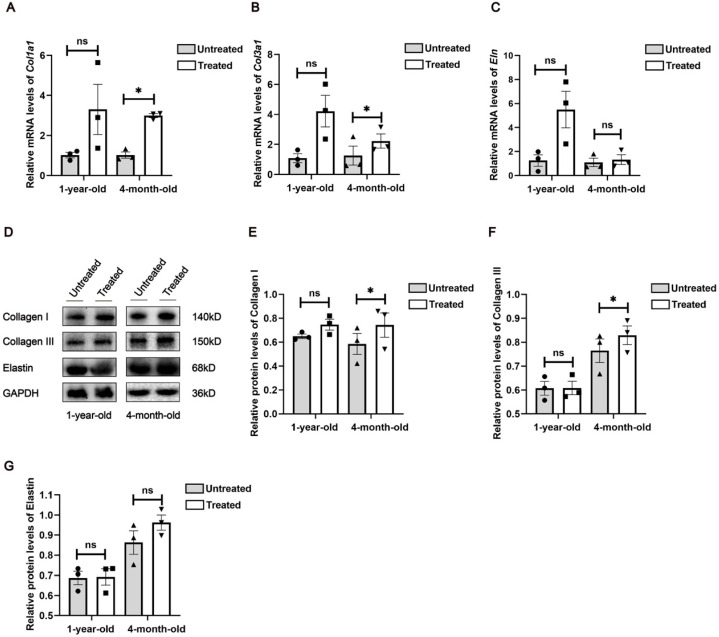
Effects of radiofrequency treatment on collagen-related genes and elastin protein at mRNA and protein levels. Real-time quantitative PCR analysis of gene expression of collagen I (**A**), collagen III (**B**) and elastin (**C**), *n* = 3. (**D**) Western blot analysis of cell lysates from mouse skin tissues. Molecular weights of target bands are indicated on the right. Protein expression was quantitated using Fiji ImageJ and normalized to internal control of GAPDH. (**E**) Collagen I. (**F**) Collagen III. (**G**) Elastin. *n* = 3. Statistical test used is paired Student’s *t*-test. * *p* < 0.05, ns: not significant.

**Figure 3 ijms-26-04460-f003:**
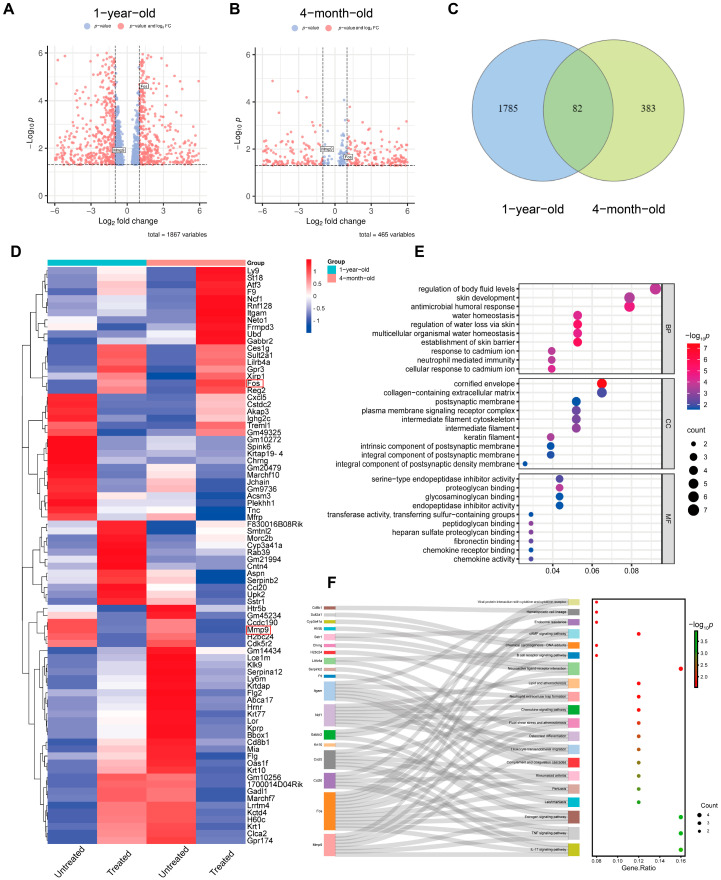
Differential expression of genes (DEGs) in mouse skin tissues treated by radiofrequencies. (**A**,**B**) Volcano plot depicting differentially expressed genes in mice of different ages. Red dots represent gene expression with a fold change ≥ 1.2 and a *p*-value < 0.05, while blue dots represent gene expression with a fold change less than 1.2. Y-axis denotes log_10_
*p*-values, while X-axis shows log_2_ fold-change values. Venn diagram (**C**) and heatmap (**D**) of DEGs. The red rectangular boxes in (**D**) highlight potential hub genes identified in subsequent analyses. Gene Ontology (**E**) and KEGG pathway enrichment analysis (**F**) of radiofrequency treatment-related DEGs. Data availability: NCBI GEO (accession number: GSE278079).

**Figure 4 ijms-26-04460-f004:**
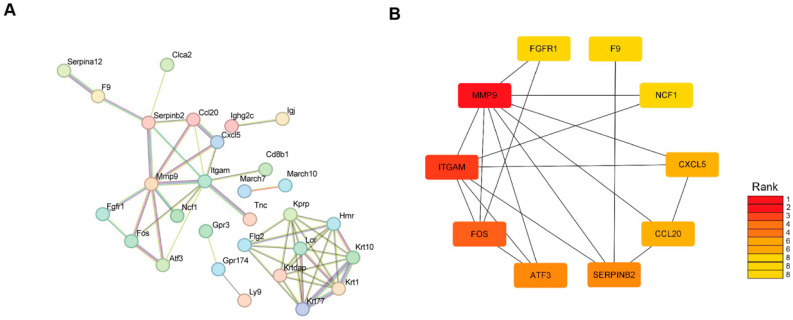
Construction of protein–protein interaction (PPI) network and identification of hub DEGs. (**A**) PPI network analysis of radiofrequency-induced differentially expressed genes (DEGs) using STRING database. (**B**) The top 10 hub DEGs with the highest connectivity identified by MCC (color depth for ranking of hub DEGs).

**Figure 5 ijms-26-04460-f005:**
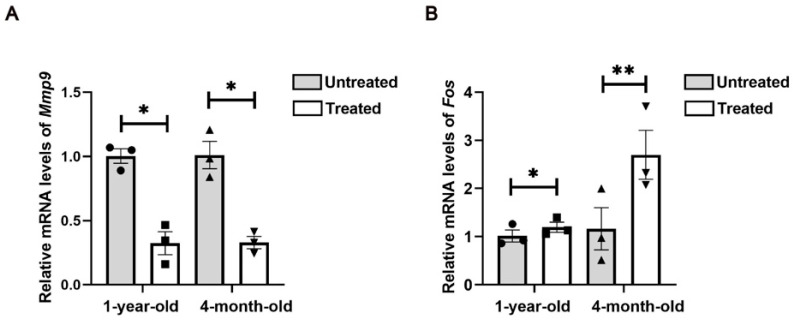
Validation of the mRNA levels of *Mmp9* (**A**) and *Fos* (**B**) by qPCR using mouse skin tissue samples. *n* = 3. Statistical test used in (**A**,**B**) is paired Student’s *t*-test. * *p* < 0.05 and ** *p* < 0.01 versus untreated.

**Figure 6 ijms-26-04460-f006:**
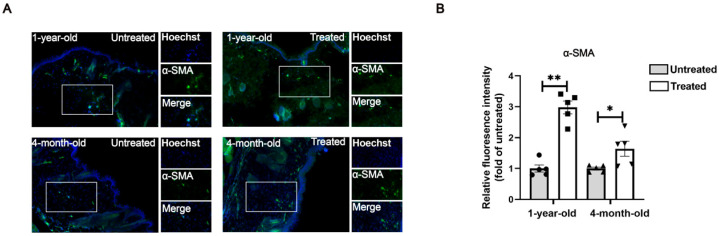
Radiofrequency treatment stimulates dermal fibroblast activation. (**A**) Representative immunofluorescent staining of α-SMA (green) in mouse skin. Hoechst (blue) counterstains nuclei. All images are representative of three independent experiments. Single-channel and marged views of the fluorescence signals are displayed for the area marked with the white box. Magnification: 400×. *n* = 5. (**B**) Quantification of fluorescence intensity using ImageJ. Statistical test used in (**B**) is paired Student’s *t*-test. * *p* < 0.05 and ** *p* < 0.01 versus untreated.

**Figure 7 ijms-26-04460-f007:**
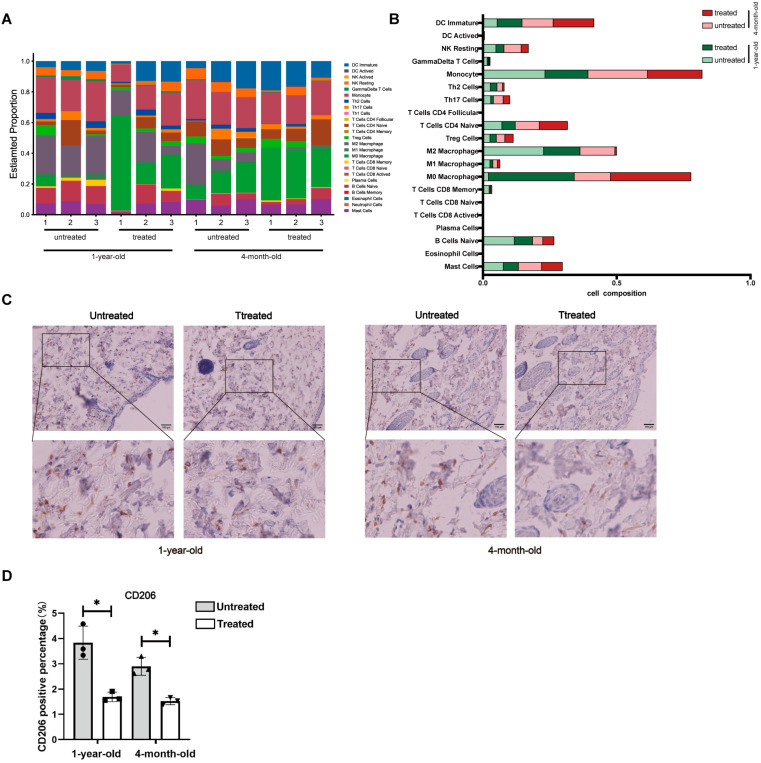
Decrease in M2 macrophages is associated with radiofrequency treatment of mouse skin. (**A**) Composition of immune cells in mouse skin tissue estimated by the CIBERSORT algorithm. Bar plot showing cell types and relative percentage of each sample. Different colors represent different cell types, which are shown in the right bar. Labels 1, 2 and 3 correspond to three independent biological replicates in each group of mice. (**B**) Bar plot showing the average relative cell composition of different groups as indicated (*n* = 3 in each group). (**C**) Representative immunohistochemistry staining of CD206 in mouse skin tissues. All images are representative of three independent experiments. Scale bar, 100 μm. (**D**) Quantification of CD206-positive cells in immunohistochemistry images taken in experiment (**C**). *n* = 3. Statistical test used in (**D**) is paired Student’s *t*-test. * *p* < 0.05 versus untreated.

**Figure 8 ijms-26-04460-f008:**
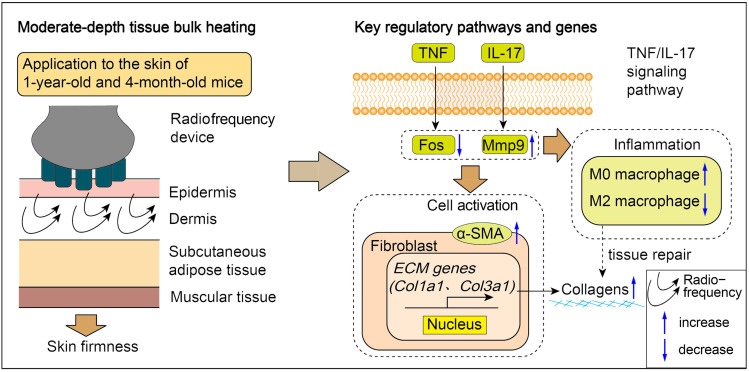
Schematic model of radiofrequency device on mouse skin with operational illustration (**left**) and potential underlying mechanisms (**right**). Radiofrequency device induces transcriptional up-regulation of Fos and down-regulation of Mmp9. These changes are associated with macrophage alternative activation and fibroblast activation, which subsequently lead to an increase in the expression of collagen-coding genes critical for maintaining skin tightness.

**Table 1 ijms-26-04460-t001:** Primer sequences used for qPCR analysis.

Gene	Forward (5′-3′)	Reverse (5′-3′)
*Col1a1*	AGACCTGTGTGTTCCCTACT	GAATCCATCGGTCATGCTCTC
*Col3a1*	GGATCTGTCCTTTGCGATGA	GTAGAAGGCTGTGGGCATATT
*Eln*	CTGCCAAAGCTGCCAAATAC	CCAACACCATAGCCAGGAAA
*Mmp9*	CTGGAACTCACACGACATCTT	TCCACCTTGTTCACCTCATTT
*Fos*	CGTCTTCCTTTGTCTTCACCTACCC	CCTTGCCTTCTCTGACTGCTCAC
*Gaphd*	AGGTCGGTGTGAACGGATTTG	TGTAGACCATGTAGTTGAGGTCA

## Data Availability

The RNA-seq data are available in the Gene Expression Omnibus (GEO) database under GSE278079.

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
