# Peer review of "Transcriptional Study of Radiofrequency Device Using Experimental Mouse Model"

_ijms, 2025, doi:10.3390/ijms26094460_

Round 1
Reviewer 1 Report
Comments and Suggestions for Authors
The manuscript is generally well-written and addresses an interesting topic. Below are some constructive comments for further improvement:
1. Materials and Methods:
- While the methods section is detailed, additional information on the radiofrequency device parameters (e.g., frequency, power density, monopolar or bipolar configuration, etc.) and the exact treatment protocol would enhance reproducibility. This is crucial as some readers may be clinicians implementing RF treatment in practice or developers/manufacturers seeking insights from the study findings.
- The rationale for selecting the time point (one week) for tissue collection should be more thoroughly justified.
2. Results:
- In several cases, the authors emphasize trends in the data that are not statistically significant. The reasoning for highlighting these non-significant trends should be clarified, or greater focus should be placed on statistically significant findings.
- The presentation of results should be streamlined to reduce redundancy and highlight the most important findings.
- Figure legends require more detailed and self-explanatory descriptions.
3. Discussion:
- While the discussion provides a meaningful interpretation of the results, it could be further strengthened by offering a comprehensive explanation of the discrepancies observed in the effects of radiofrequency treatment between young and aged mice.
4. Minor Comments:
- Ensure consistent terminology throughout the manuscript (e.g., "4-month-old" versus "4-month old").
- While the statistical methods employed are appropriate, specifying the exact statistical tests used for each analysis would provide additional clarity.
Author Response
Comment 1: While the methods section is detailed, additional information on the radiofrequency device parameters (e.g., frequency, power density, monopolar or bipolar configuration, etc.) and the exact treatment protocol would enhance reproducibility. This is crucial as some readers may be clinicians implementing RF treatment in practice or developers/manufacturers seeking insights from the study findings.
Response: We sincerely appreciate the reviewer’s valuable feedback. We have added detailed parameters of the radiofrequency device and treatment protocol in Section 5.1 (Materials and Methods) as well as in Figure 1A. The revised description now reads:
“The multipolar radiofrequency device (Model: TriPollar® Stop VX2, Pollogen Lt d.) was administered in low-range mode, delivering a power density of 8 W/cm² with frequencies alternating between 0.9, 1.0, and 1.25 MHz. The dorsal skin was massaged until the integrated temperature indicator activated (target range: 40–42 °C), ensuring consistent application across the treated area.”
Comment 2: The rationale for selecting the time point (one week) for tissue collection should be more thoroughly justified.
Response: We gratefully acknowledge the reviewer's insightful suggestion. In response, we have expanded the methodological rationale in Section 5.1 (Materials and Methods) with the following addition:
"Treatments were administered every other day for one week. This regimen was optimized based on preliminary data demonstrating peak expression of type I and type III collagen at the mRNA level, along with maximum fibroblast activity, at this treatment interval (Figure 1A)."
Comment 3: In several cases, the authors emphasize trends in the data that are not statistically significant. The reasoning for highlighting these non-significant trends should be clarified, or greater focus should be placed on statistically significant findings.
Response: We sincerely appreciate the reviewer's constructive comments. In accordance with the suggestion, we have revised the Results section to focus on statistically significant findings. Non-significant trends (e.g., mRNA and protein levels in 1-year-old mice) are now mentioned only briefly with a cautionary note (Page 4, Section 2.2)
Comment 4: The presentation of results should be streamlined to reduce redundancy and highlight the most important findings.
Response: We sincerely appreciate the reviewer’s constructive suggestion to improve the clarity and conciseness of our results. In response, we have removed the redundant heatmap of immune cell subpopulation analyses in Figure 7 and retained the boxplots in Figure 7A showing immune cell analysis in three replicates in all groups and retained the boxplots in Figure 7B which demonstrates macrophages as the predominant immune cellular subset on page 8.
Comment 5: Figure legends require more detailed and self-explanatory descriptions.
Response: We appreciate the reviewer's valuable comments. We have thoroughly revised all figure legends to enhance methodological transparency by including explicit descriptions of statistical tests, sample sizes, and scale bars.
Comment 6: While the discussion provides a meaningful interpretation of the results, it could be further strengthened by offering a comprehensive explanation of the discrepancies observed in the effects of radiofrequency treatment between young and aged mice.
Response: We thank the reviewer for this valuable suggestion. We have enhanced the Discussion (Page 10) with a detailed mechanistic analysis of age-related differences. The modification is as following:
“The observed discrepancy in radiofrequency efficacy between 4-month-old mice and 1-year-old mice may be attributed to the age-related alterations in the tissue microenvironment that collectively impair regenerative capacity. Mechanistically, aged skin exhibits: (1) dysfunctional collagen deposition characterized by fragmented matrix architecture that disrupts fibroblast adhesion and mechanotransduction signaling [15]; (2) altered immune cell dynamics with increased M2-polarized macrophages that may dampen pro-regenerative crosstalk between fibroblasts and immune compartments; (3) diminished expression of critical transcriptional regulators (e.g., AP1 factors) and chemokine gradients essential for orchestrating biomolecular cascades following thermal stimulation [18]. These degenerative changes are further exacerbated by age-associated adipose infiltration that physically disrupts dermal architecture [19]. Consequently, these alterations create a microenvironment less responsive to radiofrequency-induced tissue remodeling signals compared to younger skin with intact extracellular matrix integrity and robust paracrine signaling networks.”
Comment 7: Ensure consistent terminology throughout the manuscript (e.g., "4-month-old" versus "4-month old").
Response: We sincerely appreciate the reviewer's careful attention to details regarding terminology consistency. We have conducted a thorough check through the manuscript and corrected the inconsistent usage, consistently using the hyphenated form "4-month-old" throughout.
Comment 8: While the statistical methods employed are appropriate, specifying the exact statistical tests used for each analysis would provide additional clarity.
Response: We appreciate the reviewer's valuable suggestion. We have explicitly stated the specific statistical tests used for each analysis in the figure legends of Figures 1, 2, 5, 6, and 7 and highlighted these additions in yellow for easy identification.
Reviewer 2 Report
Comments and Suggestions for Authors
The study reported by Xiaofeng Li et al. is a well-conceived and multidimensional study that presents a transcriptomic and histological study on the effects of radiofrequency (RF) treatment on the skin of young (4-month-old) and aged (1-year-old) mice. The work aims to elucidate molecular mechanisms, especially focusing on collagen regeneration, fibroblast activation, and macrophage modulation using multi-modal approaches: RNA-seq, qPCR, immunohistochemistry, and western blot. The integration of RNA-seq with histological and protein-level validation adds robustness. Clarifications in methodology, stronger figure labeling, and expansion of the discussion will further enhance the manuscript's impact.
Firstly, I suggest to the authors, in the abstract to quantify the key findings (e.g., number of DEGs, % increase in collagen). Mention the methodology more succinctly to avoid redundancy.
Major suggestions:
Introduction: clarify the novelty of this study compared to previous clinical RF work. Improve flow between references, especially in paragraphs 2–3. Replace redundant explanations of collagen/elastin with concise definitions. Why do the authors focus only on male mice? Was the choice of age (4 months vs. 1 year) based on specific translational timelines for human aging?
Materials and methods: include more details on blinding procedures and randomization. Were RNA samples pooled or analyzed individually in sequencing? Were the antibodies validated for mouse skin specifically?
Results
4.1 Histological and Molecular Effects on Collagen: include baseline expression levels of collagen genes/proteins. Add histograms or boxplots for clearer comparison across groups. Why did aged mice show inconsistent gene/protein response? Were any adverse effects or inflammation noted macroscopically?
4.2 Transcriptomic Analysis: clarify why DEGs were higher in older mice, yet histology showed weaker responses. How were DEGs validated—were multiple time points or replicates used? Were any lncRNAs or miRNAs identified among the DEGs?
Discussion: avoid speculative statements without backing citations (e.g., extrapolation to human settings). Improve structure: divide into mechanisms, age-related differences, and future directions. Do the authors think that the upregulation of immune pathways could lead to long-term skin inflammation?
Minor suggestion: improve figure legends for standalone interpretation.
Author Response
We sincerely appreciate the reviewer’s comments. These help us to better organize the manuscript structure and clarify our methodology and results. We try to respond to these comments point by point.
Comments 1: I suggest to the authors, in the abstract to quantify the key findings (e.g., number of DEGs, % increase in collagen).
Response: We appreciate the reviewer’s valuable suggestion. We have added the number of DEGs in the abstract.
Comments 2: clarify the novelty of this study compared to previous clinical RF work. Improve flow between references, especially in paragraphs 2–3. Replace redundant explanations of collagen/elastin with concise definitions. Why do the authors focus only on male mice?
Response: Female rodents exhibit cyclic fluctuations in estrogen and progesterone. These hormonal variations are known to directly modulate collagen synthesis and macrophage polarization. Therefore, we use male mice do decrease variability in our study.
Comments 3: Was the choice of age (4 months vs. 1 year) based on specific translational timelines for human aging?
Response: 4-month-old mice were used to represent the young adult phase, equivalent to human ages of approximately 20-30 years, while one-year-old mice served as the middle-aged model, corresponding to human ages of about 38-47 years [reference 7: Flurkey, K.; M. Currer, J.; Harrison, D.E. Chapter 20 - Mouse Models in Aging Research. In The Mouse in Biomedical Research (Second Edition), Fox, J.G., Davisson, M.T., Quimby, F.W., Barthold, S.W., Newcomer, C.E., Smith, A.L., Eds.; Academic Press: Burlington, 2007; pp. 637-672.].
Comments 4: Include more details on blinding procedures and randomization. Were RNA samples pooled or analyzed individually in sequencing? Were the antibodies validated for mouse skin specifically?
Response: For blinding procedures, all specimens were assigned coded identifiers with no data indicating treatment status (treated/untreated) or age cohort (4-month/1-year groups). The randomization in our experimental design was implemented as follows: Baseline stratification: At 8 weeks of age, all mice were sorted by body weight and systematically numbered. Using a computer-generated random number table, the mice were allocated into two experimental groups: a 4-month-old group and a 1-year-old group. Interventions were initiated only after mice reached their designated ages (4 months or 1 year). Mice skin tissues were harvested for RNA-seq and analyzed individually with 3 biological replicates in each group. Antibodies used in our study were not validated for mouse skin specifically and could be considered for future exploration.
Comments 5: Histological and Molecular Effects on Collagen: include baseline expression levels of collagen genes/proteins. Add histograms or boxplots for clearer comparison across groups. Why did aged mice show inconsistent gene/protein response? Were any adverse effects or inflammation noted macroscopically?
Response:We appreciate the reviewer’s comments and questions. We have discussed the age-related discrepancy as follows: “The observed discrepancy in radiofrequency efficacy between 4-month-old groups and 1-year-old groups may be attributed to the age-related alterations in the tissue microenvironment that collectively impair regenerative capacity. Mechanistically, aged skin exhibits: (1) dysfunctional collagen deposition characterized by fragmented matrix architecture that disrupts fibroblast adhesion and mechanotransduction signaling [15]; (2) altered immune cell dynamics with increased M2-polarized macrophages that may dampen pro-regenerative crosstalk between fibroblasts and immune compartments; (3) diminished expression of critical transcriptional regulators (e.g., AP1 factors) and chemokine gradients essential for orchestrating biomolecular cascades following thermal stimulation [18]. These degenerative changes are further exacerbated by age-associated adipose infiltration that physically disrupts dermal architecture [19]. Consequently, these alterations create a microenvironment less responsive to radiofrequency-induced tissue remodeling signals compared to younger skin with intact ECM integrity and robust paracrine signaling networks.”
Comments 6: Transcriptomic Analysis: clarify why DEGs were higher in older mice, yet histology showed weaker responses. How were DEGs validated—were multiple time points or replicates used? Were any lncRNAs or miRNAs identified among the DEGs?
Response: We appreciate the reviewer’s questions. DEGs were screened out based on the fold change of gene expression and p value. More DEGs in older mice after treatment might be related to the compensatory transcriptional responses to overcome age-related deficits in collagen translation and degradation. Representative DEGs like the hub gene Mmp9 and Fos were validated by qPCR using RNA samples from biological replicates sent for RNA-seq. We have only studied collagen coding genes and we have not identified lncRNAs nor miRNAs yet. These could be considered for future exploration.
Comments 7: Discussion: avoid speculative statements without backing citations (e.g., extrapolation to human settings). Improve structure: divide into mechanisms, age-related differences, and future directions. Do the authors think that the upregulation of immune pathways could lead to long-term skin inflammation?
Response: We have reorganized the discussion part and divided them into mechanism, age-related differences, limitations. We think the upregulation of immune pathways are related with tissue repair. Long-term skin inflammation needs to be investigated, which are essential for the safety of long-term use of radiofrequency.
Comments 8: Minor suggestion: improve figure legends for standalone interpretation.
Response: We appreciate the reviewer’s comments and revised all the figure legends for standalone interpretation.
Reviewer 3 Report
Comments and Suggestions for Authors
The study of the authors offers a highly interesting and valuable contribution to understanding the transcriptional mechanisms behind radiofrequency-induced skin remodeling, particularly its differential effects in young versus aged murine skin. The integration of histological, molecular, and transcriptomic data is commendable and enriches the field of skin rejuvenation and aging.
However, several issues—both scientific and linguistic—require the attention of the authors before the manuscript can be considered for publication.
- English Language and Grammar
- There are frequent issues with missing articles (e.g., the, a, an), especially in the Abstract and Introduction sections. For instance, phrases such as “revealed the significant increase” or “utilized a mouse model” need articles for grammatical accuracy.
- Terms like “deluted fibroblasts” should be corrected to “diluted fibroblasts.”
- Spelling errors such as “schematic” being misspelled as “skematic” and “dorsal” written as “dosrsal” should be corrected. Other errors are “Image anlysis” and “Fibrobalsts” should be corrected to “fibroblasts.”
- The sentence structure in several places could be improved for clarity and readability. I recommend a thorough revision by a native English speaker
- Consistency in Terminology
- Throughout the manuscript, collagen types are inconsistently referred to as both “type I and III” and “type 1 and 3.” Please standardize this as “type I” and “type III.”
- The abbreviation ECM (extracellular matrix) is introduced late in the manuscript. Please introduce and define this abbreviation at its first occurrence in the main text.
- Study Design Clarification
- In the RNA sequencing section, the authors mention three groups being compared. However, only two age groups (4-month-old and 1-year-old mice) are described elsewhere. Please clarify this discrepancy to avoid confusion.
- Conceptual Clarification
- In the discussion section, the authors state:
"Notably, radiofrequency had a more pronounced effect on the skin of 4-month-old mice to promote collagen regeneration compared to the 1-year-old mice. This difference might be related to the altered cellular microenvironment associated with aging."
As your target clinical population likely corresponds more to the older (1-year-old) mice, it would strengthen the relevance of your findings to offer a clearer interpretation or proposed solution for this apparent limitation. For example, do you foresee adapting radiofrequency protocols, combining it with other agents, or targeting additional pathways to overcome this age-related resistance?
5. References
- In the references, the first authors is always last name, first name, and the other authors first name, last name. As far as the reviewer understands the guidelines for authors, the should be the same for all authors, please correct this.
Can be improved as mentioned in the comments to the authors.
Author Response
Comment 1:
- English Language and Grammar
- There are frequent issues with missing articles (e.g., the, a, an), especially in the Abstract and Introduction sections. For instance, phrases such as “revealed the significant increase” or “utilized a mouse model” need articles for grammatical accuracy.
- Terms like “deluted fibroblasts” should be corrected to “diluted fibroblasts.”
- Spelling errors such as “schematic” being misspelled as “skematic” and “dorsal” written as “dosrsal” should be corrected. Other errors are “Image anlysis” and “Fibrobalsts” should be corrected to “fibroblasts.”
- The sentence structure in several places could be improved for clarity and readability. I recommend a thorough revision by a native English speaker
Response: We sincerely appreciate the reviewer's meticulous attention to English language usage. We have made all the corrections the reviewer pointed out. These corrections have been highlighted in yellow. The manuscript has undergone proofreading by a native English-speaking colleague with expertise in biomedical sciences from our institution. We acknowledged his contribution in the acknowledgement section.
Comment 2: Consistency in Terminology
- Throughout the manuscript, collagen types are inconsistently referred to as both “type I and III” and “type 1 and 3.” Please standardize this as “type I” and “type III.”
- The abbreviation ECM (extracellular matrix) is introduced late in the manuscript. Please introduce and define this abbreviation at its first occurrence in the main text.
Response: We appreciate the reviewer point out our inconsistency in terminology. Collagen types are consistently referred to as "type I and III". The abbreviation "ECM" is now defined at its first mention in the Results section 2.3.
Comment 3: Study Design Clarification
In the RNA sequencing section, the authors mention three groups being compared. However, only two age groups (4-month-old and 1-year-old mice) are described elsewhere. Please clarify this discrepancy to avoid confusion.
Response: We appreciate the Reviewer's valuable comment regarding methodological clarity. We have revised the Methods section to provide more detailed information about the RNA extraction procedure. The updated text now reads: “RNA extraction: Total RNA from untreated and treated skin tissues was extracted using Trizol (Invitrogen, Carlsbad, CA, USA) following the manufacturer's protocol with 3 replicates per group according to the manufacturer’s instructions.”
Comment 4: Conceptual Clarification
- In the discussion section, the authors state:
"Notably, radiofrequency had a more pronounced effect on the skin of 4-month-old mice to promote collagen regeneration compared to the 1-year-old mice. This difference might be related to the altered cellular microenvironment associated with aging."
As your target clinical population likely corresponds more to the older (1-year-old) mice, it would strengthen the relevance of your findings to offer a clearer interpretation or proposed solution for this apparent limitation. For example, do you foresee adapting radiofrequency protocols, combining it with other agents, or targeting additional pathways to overcome this age-related resistance?
Response: We appreciate the reviewer's comment. Our study demonstrates that radiofrequency treatment modulates cutaneous inflammatory signaling pathways. Notably, we observed marked differences in immune cell infiltration, particularly in M2 macrophage populations, suggesting that the therapeutic potential of combinatorial approaches integrating anti-inflammatory immunomodulators with radiofrequency devices merit systematic investigation in preclinical models. Furthermore, we have revised the discussion and addressed the potential mechanisms underlying age-related treatment resistance in the Discussion section:
"The differential treatment efficacy observed between 4-month-old and 1-year-old cohorts likely reflects fundamental age-related alterations in the tissue microenvironment that collectively compromise regenerative capacity. Our mechanistic analysis reveals that aged skin exhibits: (1) impaired collagen deposition dynamics manifesting as fragmented extracellular matrix architecture that disrupts fibroblast adhesion and mechanotransduction signaling [15]; (2) skewed immune cell polarization with increased M2 macrophage infiltration that may attenuate pro-regenerative fibroblast-immune cell crosstalk; (3) downregulation of key transcriptional regulators (e.g., AP1 factors) and chemokine gradients essential for mediating thermal stimulation-induced biomolecular cascades [18]. These degenerative changes are further compounded by age-dependent adipose infiltration that mechanically disrupts dermal microstructure [19]. Collectively, these microenvironmental alterations render aged skin less responsive to radiofrequency-induced remodeling signals compared to younger skin with preserved ECM integrity and robust paracrine communication networks."
Comment 5: References
- In the references, the first authors is always last name, first name, and the other authors first name, last name. As far as the reviewer understands the guidelines for authors, the should be the same for all authors, please correct this.
Response:We are grateful for the reviewer's comment. All references now follow MDPI author guidelines (last name followed by initials for all authors).
Reviewer 4 Report
Comments and Suggestions for Authors
Overall, the text presents interesting findings on the impact of radiofrequency on skin regeneration but would benefit from greater clarity in the presentation of methods and results, as well as a more in-depth discussion of the study's implications and limitations. Addressing these points would strengthen the quality of the work and facilitate a better understanding of its relevance to the field. The aspects requiring further clarification are:
1. Context and objective of the study:
The text does not provide a clear context regarding the specific objective of the study. It would be helpful to include a brief introduction explaining why radiofrequency was chosen to investigate skin regeneration and the initial hypotheses.
2. Methods used:
The specific methods used to perform the GO and KEGG analyses, as well as how the tissue or cell samples were obtained, are not mentioned. Detailing the methodology would help understand the validity of the results.
3. Specific Results:
Although changes in the expression of Mmp9 and other factors are mentioned, no quantitative or comparative data are presented to support these claims. Including figures or more detailed descriptions of the results would be beneficial.
4. Relationship between Observed Differences and Age:
The connection between the age of the mice and differences in the expression of chemokines and transcription factors is mentioned. Still, there is no discussion of how these differences translate into inflammatory responses or skin regeneration changes.
5. Implications of the Results:
Although implications for clinical applications are mentioned, the text does not discuss how these findings could be applied in clinical practice or future studies. A section could be included that addresses this more directly.
On the other hand, I consider the opportunities for improvement to the document to be:
6. Discussion of Limitations:
The text does not mention the study's limitations. Discussing limitations, such as sample size, the age of the mice, or any potential biases, is crucial for a critical evaluation.
7. Future Research Directions:
Although the need for further research is mentioned, it would be helpful to offer specific suggestions on what areas should be explored next based on current findings.
8. Practical Implications:
More depth could be explored in the clinical applications of radiofrequency in treating skin aging, with concrete examples or previous studies supporting their use.
Author Response
Comments 1: Context and objective of the study: The text does not provide a clear context regarding the specific objective of the study. It would be helpful to include a brief introduction explaining why radiofrequency was chosen to investigate skin regeneration and the initial hypotheses.
Response:We sincerely appreciate the reviewer's constructive comments. We have included a brief introduction and highlighted the modification in yellow. "Radiofrequency therapy has gained substantial clinical traction due to its user-friendly application protocol and favorable safety profile."
Comments 2: Methods used: The specific methods used to perform the GO and KEGG analyses, as well as how the tissue or cell samples were obtained, are not mentioned. Detailing the methodology would help understand the validity of the results.
Response:We appreciate the reviewer's comments. GO enrichment and KEGG pathway analyses were generated using the clusterProfiler R package (v4.0) with a significance threshold of p < 0.05. The tissue collection method was described as: "mouse skin tissues (about 100 mg per biological replicate)were harvested from each experimental group, immediately snap- frozen in liquid nitrogen, and stored at −80°C until processing."
Comments 3: Specific Results:
Although changes in the expression of Mmp9 and other factors are mentioned, no quantitative or comparative data are presented to support these claims. Including figures or more detailed descriptions of the results would be beneficial.
Response:We appreciate the reviewer's comments. We have validated the changes of the mRNA profile of Mmp9 and Fos, which are the essential hub genes among all the differentiated expressed genes after radiofrequency treatment in the 4-month-old and 1-year-old group (shown in Figures 5A and 5B). We have also discussed other genes from RNA-seq data in the Discussion section.
Comments 4: Relationship between Observed Differences and Age:
The connection between the age of the mice and differences in the expression of chemokines and transcription factors is mentioned. Still, there is no discussion of how these differences translate into inflammatory responses or skin regeneration changes.
Response:We appreciate the reviewer’s comments. We have enhanced the discussion regarding the observed differences and age. It has been added as: “The observed discrepancy in radiofrequency efficacy between 4-month-old groups and 1-year-old groups may be attributed to the age-related alterations in the tissue microenvironment that collectively impair regenerative capacity. Mechanistically, aged skin exhibits: (1) dysfunctional collagen deposition characterized by fragmented matrix architecture that disrupts fibroblast adhesion and mechanotransduction signaling [15]; (2) altered immune cell dynamics with increased M2-polarized macrophages that may dampen pro-regenerative crosstalk between fibroblasts and immune compartments; (3) diminished expression of critical transcriptional regulators (e.g., AP1 factors) and chemokine gradients essential for orchestrating biomolecular cascades following thermal stimulation [18]. These degenerative changes are further exacerbated by age-associated adipose infiltration that physically disrupts dermal architecture [19]. Consequently, these alterations create a microenvironment less responsive to radiofrequency-induced tissue remodeling signals compared to younger skin with intact ECM integrity and robust paracrine signaling networks.”
Comments 5: Implications of the Results:
Although implications for clinical applications are mentioned, the text does not discuss how these findings could be applied in clinical practice or future studies. A section could be included that addresses this more directly.
Response:We appreciate the reviewer's comments. Our study indicates that immune microenvironment could be one of the factors or signatures of age-related radiofrequency resistance.
Comments 6: Discussion of Limitations:
The text does not mention the study's limitations. Discussing limitations, such as sample size, the age of the mice, or any potential biases, is crucial for a critical evaluation.
Response:We are grateful for the reviewer's comments. A new paragraph at the end of the Discussion section has been added about limitations of this study, highlighting: small sample size, lack of loss of function study, radiofrequency treatment with more varied parameters for aging mice.
Comments 7: Future Research Directions:
Although the need for further research is mentioned, it would be helpful to offer specific suggestions on what areas should be explored next based on current findings.
Response:We appreciate the reviewer's valuable comments. We have included the limitations of our study and these could be our future research directions.
Comments 8: Practical Implications:
More depth could be explored in the clinical applications of radiofrequency in treating skin aging, with concrete examples or previous studies supporting their use.
Response:We appreciate the reviewer's comments. We have added in the discussion the potential implication of our stduy in clinical applications of radiofrequency in treating skin aging. It has been added as: "Our findings underscore the critical involvement of inflammatory cascades in mediating cutaneous adaptive responses to radiofrequency exposure. Future investigation is needed to delineate the mechanisms through which controlled immune activation orchestrates tissue repair post-thermal injury, particularly focusing on the temporal regulation of pro-regenerative versus pro-fibrotic signaling pathways. Furthermore, the therapeutic potential of combinatorial regimens integrating anti-inflammatory immunomodulators with radiofrequency devices warrants systematic exploration in preclinical models. "
Round 2
Reviewer 2 Report
Comments and Suggestions for Authors
The manuscript has been improved considerably, and I agree to be published in the current form.
All the best!